# Ozone Infiltration for Osteonecrosis of the Jaw Therapy: A Case Series

**DOI:** 10.3390/jcm11185307

**Published:** 2022-09-09

**Authors:** Olga Di Fede, Carmine Del Gaizo, Vera Panzarella, Gaetano La Mantia, Pietro Tozzo, Anna Di Grigoli, Antonio Lo Casto, Rodolfo Mauceri, Giuseppina Campisi

**Affiliations:** 1Department of Surgical, Oncological and Oral Sciences, University of Palermo, Via del Vespro 129, 90127 Palermo, Italy; 2Unit of Stomatology, Azienda Ospedaliera Ospedali Riuniti “Villa Sofia-Cervello” of Palermo, Piazza Salerno, 1, 90146 Palermo, Italy; 3Private Practice, Via Accardo 4, 90145 Palermo, Italy; 4Biomedicine, Neuroscience and Advanced Diagnostic Department, University of Palermo, Via del Vespro 129, 90127 Palermo, Italy; 5Unit of Oral Medicine and Dentistry for Fragile Patients, Department of Rehabilitation, Fragility and Continuity of Care, University Hospital Palermo, Via del Vespro 129, 90127 Palermo, Italy

**Keywords:** osteonecrosis of the jaw, ONJ, ozone, treatment, conservative, healing, imaging

## Abstract

Medication-related osteonecrosis of the jaw (MRONJ) is a serious adverse reaction, mainly due to bone-modifying agents (BMA), and it is a potentially painful and debilitating condition. To date, the literature has reported a 90% rate of successful outcomes for MRONJ patients undergoing surgical treatment. Particularly for patients with advanced disease stages who are unsuitable for surgery, prolonged medical treatment is required, with a consequent risk of the overuse of antibiotics and antibiotic resistance. The aim of this study is to evaluate the efficiency and safety of ozone, via oral mucosal infiltrations, in seven cancer patients with MRONJ, who are not eligible for surgery. The protocol (OZOPROMAF) consists of intratissue injections of an oxygen ozone (O_2_O_3_) mixture, which is applied until formation of a sequestrum and clinical healing. Follow-up was scheduled to confirm the healing of MRONJ and radiological evaluations by CBCT were planned. In order to assess the level of pain, a questionnaire including the Numeric Rating Scale for Pain (NRS Pain) was administered on the first visit, one day after treatment, and one week after treatment. After an application of OZOPROMAF, all patients reported discomfort for some hours, probably due to soft tissue pressure around the infiltration site. Thereafter, the discomfort subsided within 6–8 h. Complete mucosal healing of MRONJ occurred within a number of cycles ranging from 7 to 16. Complete resolution with an improvement in bone condition was observed in all patients. The MRONJ lesions of all patients healed after 18–24 months. The authors of this study contend that these preliminary results suggest the efficiency and safety of the O_2_O_3_ mixture. However, further research is required to confirm the efficacy of the O_2_O_3_ mixtures in MRONJ treatment, at least for patients who are unsuitable for surgery.

## 1. Introduction

Medication-related osteonecrosis of the jaws (MRONJ) can be defined as “an adverse reaction, which is characterized by the progressive destruction and necrosis of the mandibular and/or maxillary bone, in subjects exposed to treatment with drugs with an established increased risk of disease, in the absence of previous radiation treatment” [1].

The treatment of MRONJ is challenging, and an effective and appropriate therapy that substantially improves the outcome remains to be identified [2]. According to the American Association of Oral and Maxillofacial Surgeons (AAOMS), emphasis is placed on the acceptability of nonoperative (or medical) and operative (or surgical) management for all stages of this disease. This issue is based on surgical judgment and patient-related factors in a shared decision-making model [3]. The aim of every MRONJ therapeutical protocol should be to control infection and pain, and to reduce the extent of lesions and development of new areas of necrosis [4,5]. Clinically, the lesions in MRONJ patients can heal (i.e., the maintenance of mucosal coverage is seamless and symptomless) [6], while the bone area, in which MRONJ can develop and recur, may show demarcation of the necrotic process without *restitutio ad integrum* [7]. In addition, in order to exclude a recurrence at 6–12 months after MRONJ treatment [8], clinical healing and demarcation of necrosis must be maintained. Unfortunately, the literature often reports very short follow-up periods of approximately 6 months [7].

When MRONJ is diagnosed, its stage must be determined accurately, and the treatment standardized in accordance with medical guidelines. Personalized adjustments must be made, taking into consideration the patient’s medical history [9]. Whereas significant comorbidities can preclude operative treatment, nonoperative strategies have been proposed for all stages, even if there is a paucity of homogeneous results in the literature. Furthermore, when operative treatments are suitable for the patients, management could be supported by nonoperative treatment, to be applied before or after surgery as adjuvant or neo-adjuvant protocols. Such procedures have demonstrated differing results, negative outcomes, or an improvement in signs and/or symptoms to manage, in the absence of complete healing [8].

Of those nonoperative procedures, the use of medical ozone (O_3_) has been deployed and evaluated on account of its chemical properties. O_3_ has generally been proven to play a role in the treatment of chronic, non-healing, or ischemic wounds due to its antimicrobial and anti-oxidant properties and to bio-stimulation. It has also been extensively used in different medical approaches and for different purposes [10]. Local applications are performed by a transcutaneous O_3_ gas bath for the treatment of external wounds, by tissue infiltration (e.g., intramuscular) for musculoskeletal disorders, and by ozonized water (i.e., spray or compress) or gel for oral diseases [11]. To date, insufflation or an oil suspension of medical O_3_ have been used as a surgical adjuvant for treating MRONJ with promising results [12,13,14,15,16,17]. The aim of this study is to evaluate the efficiency and safety of a mixture of O_2_O_3_ and an innovative method of application, that is, infiltrations into the oral mucosa of MRONJ patients who are ineligible for a surgical standard of care.

## 2. Materials and Methods

### 2.1. Setting

This study, entitled the OZOPROMAF protocol, was reviewed and approved by the Ethics Committee of University Hospital of Palermo, Policlinico P. Giaccone (approval number 01/2018). Patients were consecutively enrolled from February 2018 to March 2020. The study was also registered ClinicalTrials.gov (identifier: NCT05036837, 1 January 2017). The study protocol conformed with the ethical guidelines of the 1964 Declaration of Helsinki and its later amendments or comparable ethical standards. All participants gave their written informed consent. The study was performed according to the STROBE Statement.

### 2.2. Partecipants

According to the considered criteria of inclusion and exclusion, the study regarded only MRONJ cases, and they were reported to the AIFA, the Italian Medicines Agency, for the registration of adverse drug-related events. 

The inclusion and exclusion criteria are described in Table 1.

Seven cancer patients (five males/two females, mean age 67.5 years), consecutively recruited, were included in the study. Four patients were declined surgical protocols due to an elevated degree of invasiveness, and three patients could not be considered due to their severe anesthesiologic risk (ASA III/IV).

### 2.3. Variables and Data Sources

The medical, pharmacological, and dental history of patients were recorded during the initial examination (T0). Specifically, the following data were collected: (1) age; (2) sex; (3) indications for use of MRONJ-related drugs (BMA and others); (4) type and duration of MRONJ-related medication use; (5) cumulative dose, where applicable; (6) medical history of chemotherapy; (7) concurrent use of other medications, including the use of steroids; (8) other concomitant diseases; and (9) smoking habit. All patients discontinued MRONJ-related drugs after an MRONJ diagnosis, in accordance with specialist advice. All patient data are reported in Table 2. 

On the first visit (T0) and in the presence of an abscess, two of seven patients were treated once with *per os* antibiotics (i.e., amoxicillin clavulanate, 1 g, 3 times per day, and metronidazole 250 mg, twice a day) and local antiseptics (i.e., chlorhexidine 0.2% mouthwash); this procedure lasted for 7 days. A written informed consent form, acknowledging the off-label use of metronidazole in Italy, was requested. Thereafter, a maxillofacial cone beam computed tomography (CBCT) or multi-slice computed tomography (MSCT) scan was performed to ascertain staging, according to Bedogni et al. [19] at the time of the MRONJ diagnosis. The treatment outcomes were divided into clinical (complete healing, partial healing, stable disease, and progressive disease) [20] and radiological (improvement, no change, and progression) [21].

Of note, an MRONJ extension was defined not only from the overall size of the lesion site but also from an identification of the presence of MRONJ, that is, if it was limited to the alveolar bone (local disease) or if it also involved the basal bone (diffuse disease), according to SICMF-SIPMO staging [4]. There were more MRONJ lesions in three patients, and the protocol was only performed at one site where the patients reported more pain regarding mastication and eating.

### 2.4. Intervention and Quantitative Variables

The OZOPROMAF protocol consists of a local superficial anesthesia by applying an EMLA^®^ cream, and intra-tissue injections of a 15 mL OxigenOzone (O_2_O_3_) mixture with a 26Gx 1⁄2—0.45 × 13 mm needle into the mucosal margin, surrounding the bone exposure or around the site, which had previously been highlighted by a CBCT scan. OZOPROMAF was applied for a 7- to 15-day period, depending on patient compliance and availability, until the formation of sequestrum and clinical healing (T1). The pain intensity was assessed at each visit by means of a questionnaire. The latter contained items relating to pain and/or other symptom evaluation, prior to the procedure and based on a numerical rating scale (NRS). Each patient was also supported by means of a telephone call one day after the ozone procedure, in which pain and/or adverse events arising from the questionnaire were evaluated. Where the patient was unavailable for their weekly treatment, a telephone call enquiring about the questionnaire was obligatory in order to evaluate symptoms by asking simple questions and using the NRS scale. 

After the formation of sequestrum and clinical healing (T1), follow-up visits were scheduled to confirm the healing of MRONJ at 1 (T2), 3, 6 (T3), 12 (T4), and 18–24 months (T5). Following the OZOPROMAF protocol, follow-up visits were scheduled at 1 (T2), 3, 6 (T3), 12 (T4), and 18–24 months (T5) (Figure 1 and Figure 2).

An orthopantomography was performed at T3 for every patient in order to evaluate lesion stability/development and to consider radiation protection issues; a maxillofacial CBCT or MSCT scan was also performed at T4. Positive outcomes for the OZOPROMAF protocol were evaluated at T1 as there were no clinical signs of acute phlogosis and no symptoms, which were compatible with MRONJ. At the most recent follow-up visit (T5), all patients had intact mucosa and no clinical or radiological signs of MRONJ.

### 2.5. Statistical Methods

The statistical units are the patients who satisfy the inclusion criteria of the study. The descriptive statistics of all data were conventionally calculated. Statistical analysis was performed using Stata/SE 14.1 (Stata Corporation, College Station, TX, USA), and an alpha value of 0.05 was considered significant.

## 3. Results

All the data relating to the patients are summarized in Table 3.

On the first visit (T0), patients were asked to rate their pain on a single unidimensional pain scale (NRS). All patients reported significant pain with a range between 6 and 10 at T0. Two patients received applications every week, and five patients received applications every 15 days, depending on their level of compliance or availability. At every application, all OZOPROMAF patients reported discomfort, which was probably due to the pressure of the soft tissues surrounding the infiltration site. This persisted for a few hours after the procedure, being alleviated after a maximum of 6 h. No other transitory or permanent side effects were reported. Pain relief was reported to be pronounced after administering O_2_O_3_; 7/7 patients reported a reduction in postoperative pain from >8 to <3 on the NRS scale, one week after the first application of O_2_O_3_. Every patient was evaluated 24 h after treatment and after 7 days, but no collateral effects or pain in the injection zone were reported. The complete healing of MRONJ (T1) occurred after a number of cycles ranging from 7 to 16; the time frame from the commencement of treatment with O_2_O_3_ and the identification of bone sequestration ranged from 2 to 4 months. The frequency and timing of administration are reported in Table 4. 

Bony sequestrum of the necrotic bone and spontaneous expulsion were reported for all patients, and no further surgical and/or medical approach for MRONJ was required.

CBCT confirmed an appreciable improvement in the diminution of MRONJ for all sites, which had been treated with O_2_O_3_ at T4, and it was well-defined at T5. Specifically, the disappearance of ground glass in many cases at T4 and signs of bone formation were evident in six out of seven cases. CBCT confirmed the stability of MRONJ at T5 with pronounced bone healing. Significant bone improvement was observed in two cases at T4, and a case of complete bone healing at T5 (Table 4). None of the seven patients displayed residual bone lesions after treatment. Two patients could not be evaluated at T5 due to a progression of the primary disease and subsequent death. Five patients presented clinical healing and an improvement in bone healing, as confirmed by CT scans, until enhanced bone apposition.

## 4. Discussion

To date, MRONJ management has included different approaches, ranging from nonoperative (i.e., conservative and medical treatment) to operative (i.e., surgery) procedures: conservative and medical treatment (analgesic for pain relief, antibiotics, antiseptics), minimally invasive surgical treatment (e.g., curettage or debridement of the exposed area, contouring of sharp bony edges, sequestrectomy), or invasive–extensive surgical treatment (e.g., marginal and segmental resection with reconstruction of defective bone and soft tissues) [4,22]. MRONJ treatments can be defined as effective in cases of complete resolution and improvement, and no response in cases of disease stabilized and exacerbation, considering the following possible outcomes: (I) complete resolution; (II) improvement; (III) disease stabilized; (IV) exacerbation [21].

Even if MRONJ protocols could warrant functional rehabilitation with clinical healing and pain relief, in no case is organic bone recovery achieved. Thus, innovative research for more appropriate protocols will be pursued until a gold standard is reached. Nowadays, positive outcomes and features of the healing of MRONJ can be considered as the absence of phlogosis and the demarcation of the necrotic process [23]. Many treatments and protocols that are associated with adjuvant therapies (e.g., PRP, laser, BMP, teriparatide therapy, ozone therapy, oxygen therapy, and photodynamic therapy) have been proposed in the medical literature. The aim of these is to promote, to a greater or lesser degree, the healing of bone and soft tissues [4,7,24,25,26,27,28,29,30,31,32,33].

Medical ozone (O_3_) has excellent chemical properties, which are performed by: (i) activating cellular metabolism; (ii) reducing pro-inflammatory prostaglandins synthesis or the release of algogenic compounds; (iii) increasing the release of immune-suppressor cytokines; (iv) reducing oxidative stress through induction of the synthesis of antioxidant enzymes (superoxide dismutase, glutathione peroxidase, and catalase); and (v) improving the supply of tissue O_2_ by means of hemorheological action, vasodilatation, and angiogenesis stimulation [10,11]. O_3_ has also been proven to play a role in the treatment of chronic, non-healing, or ischemic wounds due to its antimicrobial and antioxidant properties and to bio-stimulation. These effects derive from its ability to induce marked oxidative stress, which stimulates cell protective mechanisms; damage the bacterial cell envelope and viral capsid; inhibit fungi growth; and interfere with the reproductive cycle (via processes of peroxidation), thereby disrupting virus-to-cell contact. Moreover, marked oxidative stress leads to increased oxygen release by the red blood cells, thereby stimulating the glycolysis rate, production of ATP, and Krebs cycle. It enhances red blood cell concentration and the hemoglobin rate, diapedesis, and phagocytosis, thereby invigorating the reticulo-histiocyte system. This is in addition to assisting the production of prostacyclin (a vasodilator) and enzymes, acting as free radical scavengers and cell-wall protectors: glutathione peroxidase, catalase, and superoxide dismutase. Finally, this marked oxidative stress increases the production of interferon, which is a tumor necrosis factor, and interleukin-2, thereby activating the immune system [10]. 

In the field of dentistry, ozone has been studied as a potential cure or adjuvant in patients with MRONJ. For example, in a study by Ripamonti et al., a special bell-shaped insufflation device was used to administer local O_3_ by insufflation in 24 oncological patients, who were affected by stage two zoledronic acid-related ONJ, every third day for a minimum of 10 days for each pathological area of disease. Of these patients, 18 had sequestrum and complete or partial spontaneous expulsion of the necrotic bone, followed by oral mucosa reepithelization after a range of 4–38 of O_3_ gas insufflations, reporting no adverse events [12]. 

Another study by Ripamonti et al. reported data regarding 10 cancer patients affected by MRONJ lesions, which were treated with 10 local applications (once every 3 days) of medical O_3_, delivered in an oil suspension for 10 min. In all patients, the mucosal lesions healed with 3–10 applications of medical O_3_ without surgical intervention, completely reconstituting oral and jaw tissue in the absence of toxicity. Moreover, new bone formation around the necrotic area was observed in two patients [15]. Petrucci et al. and Grillo et al. used O_3_ as an adjuvant treatment or in combination with surgical therapy in order to reduce the concomitant local complications of MRONJ, or to prevent the occurrence of MRONJ in cancer patients treated with bisphosphonates [14].

In all the aforementioned papers, O_3_ was administered via topical applications, with minimal discomfort, even if the results were not always satisfactory either for patients or clinicians. Thus, the challenge of clinicians in treating MRONJ patients is undoubtedly to select the most appropriate medical protocols in maximizing a positive outcome for the patient. In addition, especially regarding cancer patients and those affected by systemic life-threatening diseases, patient-centered care should be optimized since guidelines and recommendations cannot always be applied due to the elevated risk of surgical procedures and possible poor quality of life outcomes for those patients undergoing cancer treatment. Indeed, the authors of this study do not know of any academic papers considering which treatment is more appropriate for treating MRONJ on the basis of systemic conditions of the patient.

In the study outlined in this paper, the long-term effectiveness of direct intra-epithelial O_2_O_3_ injection therapy in seven cancer patients with MRONJ, otherwise not surgically treatable, was observed. In all cases, O_2_O_3_ infiltrations provided positive effects for patients in the short- and long-term, leading to spontaneous sequestrum, mucosal healing, diminished pain, and the avoidance of systemic antimicrobial treatments over extended periods of time. These infiltrations afforded greater protection to the MRONJ patient from the abuse of antibiotics, thereby addressing a key clinical issue [11].

As described in Table 4, the CBCT findings of OZOPROMAF patients have demonstrated varying degrees of MRONJ improvement, ranging from a stabilized disease, post spontaneous sequestrum, to bone healing. It can be stated that positive results have been obtained in the field of MRONJ treatment on several occasions and during the extended follow-up period. This is also the case regarding radiological findings, considering the longer bone healing time, compared with mucosal changes in response to O_2_O_3_ therapy [34].

Of note, in patients with multiple MRONJ sites, only one site for each subject was treated with OZOPROMAF, and a marked clinical and radiological improvement was observed; in contrast, other MRONJ sites had deteriorated.

During the follow-up period, two patients died due to the progression of metastases related to their primary disease (breast and prostate cancer, respectively). While it was not possible to evaluate their conditions at T5, these two patients showed clinical healing at the latest follow-up (T4), and they made no mention of suffering from pain. Thus, it can be deduced that, albeit for a short period of time, their quality of life had improved.

Considering the limitations of this present study (a small sample size, monocentric study, and heterogeneous drug administration, open type), further studies will be required to confirm whether the use of O_2_O_3_ could significantly improve clinical and bone healing in patients with MRONJ, especially regarding those patients unsuitable for surgery.

## 5. Conclusions

The authors have hypothesized that O_2_O_3_ injections could be an innovative, powerful, and effective tool for nonoperative MRONJ treatments, especially when operative (invasive) procedures could be an additional burden for complicated cases and medical challenges. During an extended follow-up period, the optimal healing of MRONJ lesions in the seven cancer patients has been observed. Interestingly, in some cases, bone remodeling was also noted. OZOPROMAF can, therefore, be considered a convincing nonoperative therapy for promoting the sequestrum of the necrotic bone, accelerating wound healing and inducing bone formation.

## Figures and Tables

**Figure 1 jcm-11-05307-f001:**
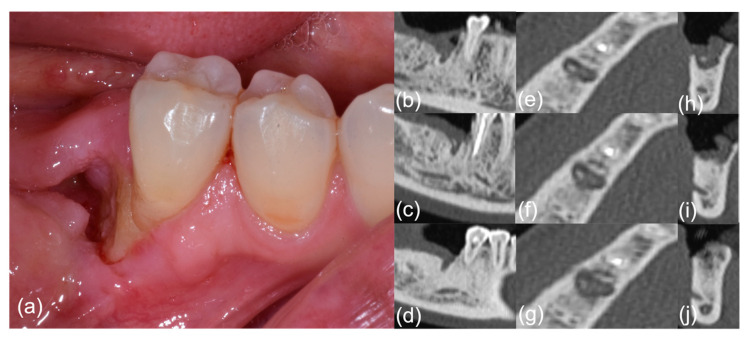
Patient #6, pre-treatment phase; right-side mandibular MRONJ: (**a**) clinical view; (**b**–**j**) computed tomography scan sections.

**Figure 2 jcm-11-05307-f002:**
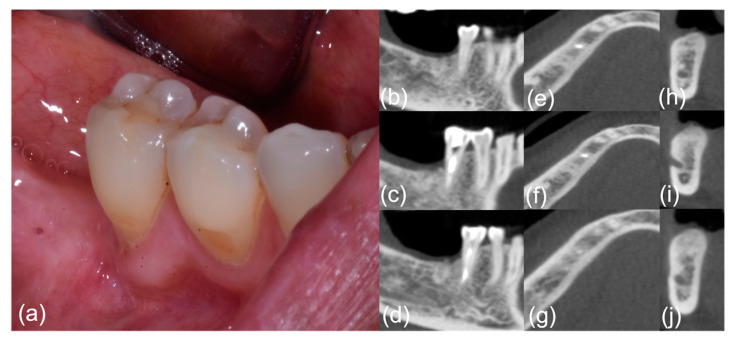
Patient #6, post-treatment phase; right-side mandibular MRONJ: (**a**) clinical view; (**b**–**j**) computed tomography scan sections.

**Table 1 jcm-11-05307-t001:** Inclusion and exclusion criteria.

**Inclusion criteria**	age ≥ 18 years clinical-radiological diagnosis of MRONJ, according to SICMF-SIPMO (all stages) [18] current ineligibility for protocols (conservative/medical for long term, or only surgical, as proposed by SICMF-SIPMO task force experts [4] regarding unstable systemic conditions, or unaccepted consent due to an invasive approach, in turn due to an advanced disease stage)
**Exclusion criteria**	previous radiation in the head and neck area neoplastic involvement of the jaw previous long-term conservative/medical or surgical treatment for MRONJ

**Table 2 jcm-11-05307-t002:** Descriptive data of enrolled patients.

Patient	Age	Sex	Primary Disease	Type of Administered ONJ-Related Drugs	Duration (Months)	Cumulative Dose	Chemo-Therapy	Other Medications	Other Diseases	Smoking Habit	MRONJ Location
**#1**	52	F	MTS/BC	Bevacizumab	4	4800 mg	No	No	No	No	Maxilla (single–right side)
**#2**	74	M	MTS/PC	Denosumab	8	60 mg	Yes	Prednisone	Hypertension	No	Mandibula (single–right side)
**#3**	73	M	MM	Zoledronate	7	5 mg	No	Prednisone	Hypertension and ischemic cardiomyopathy	Yes	Mandibula (single–right side)
**#4**	49	M	MTS/PC	Denosumab	36	360 mg	Yes	No	No	Yes	Maxilla and mandibula (multiple–right maxilla and left mandibula)
**#5**	48	M	MTS/LC	Denosumab	12	120 mg	No	Nintedanib	No	Yes	Maxilla and mandibula (multiple–right maxilla and left mandibula)
**#6**	46	F	MTS/BC	Ibandronate Denosumab Everolimus	1 38 39	3 mg 360 mg 11,700 mg	Yes	No	No	No	Mandibula (single–left side)
**#7**	83	M	MTS/PC	Zoledronate Denosumab	168 24	70 mg 180 mg	Yes	No	Atherosclerosis	No	Maxilla and mandibula (multiple–left maxilla and mandibula)

Legend: MM = multiple myeloma; MTS/BC = metastasis/breast cancer; MTS/PC = metastasis/prostate cancer; MTS/LC = metastasis from lung cancer;

**Table 3 jcm-11-05307-t003:** Detailed results of patients treated with OZOPROMAF protocol.

Patient	T0			OZOPROMAF Applications		T1	T2-T3-T4	T5
	Clinical Signs and Symptoms	NRS for Pain	MRONJ Staging *	No. O_2_O_3_ Administrations/Week	O_3_ Cycles	No. Weeks before Auto Sequestrum	Clinical Signs and Symptoms	
**#1**	**E** **Pu** **P**	8	II	1	15	16	Complete healing—no symptoms	Dead
**#2**	**E** **Pu** **P**	**7**	II	2	8	5	Complete healing—no symptoms	Complete healing
**#3**	**Pu** **S** **Pa** **P**	**10**	II	1	10	11	Complete healing—no symptoms	Complete healing
**#4**	**A** **E** **P**	**10**	II	2	15	8	Complete healing—no symptoms	Complete healing
**#5**	**F** **E** **P**	**7**	II	2	8	5	Complete healing—no symptoms	Complete healing
**#6**	**De** **E** **P**	**6**	II	2	13	7	Complete healing—no symptoms	Complete healing
**#7**	**F** **E** **D**	**7**	II	2	9	5	Complete healing—no symptoms	Dead

Legend: A = abscess; P = pain; De = mucosal dehiscence; E = exposed bone; F = fistula; Pa = paresthesia; Pu = purulent discharge; S = swelling; * according to Bedogni et al. [19].

**Table 4 jcm-11-05307-t004:** Details of CT/CBCT findings.

Patient	T0 Clinical Signs/Symptoms	T0 Pain -NRS Scale	T0 MRONJ Staging *	T4 CT Findings/Outcome	T4 Clinical Signs/Symptoms	T5 CT Findings/Outcome	T5 Clinical Signs/Symptoms
**#1**	**E** **Pu** **P**	**8**	II	Ground glass disappearance (improvement)	No pain/no ONJ signs	n.a.	n.a.
**#2**	**E** **Pu** **P**	**7**	II	Bone healing (improvement)	No pain/no ONJ signs	Stable (improvement)	No pain/no ONJ signs
**#3**	**Pu** **S** **Pa** **P**	**10**	II	Slightly improved bone formation with persistent ground glass (improvement)	No pain/no ONJ signs	Bone formation (improvement)	No pain/no ONJ signs
**#4**	**A** **E** **P**	**10**	II	Slightly improved bone formation (improvement)	No pain/no ONJ signs	Periosteal reaction (improvement)	No pain/no ONJ signs
**#5**	**F** **E** **P**	**7**	II	Slightly improved bone formation (improvement)	No pain/no ONJ signs	Stable (improvement)	No pain/no ONJ signs
**#6**	**De** **E** **P**	**6**	II	Slightly improved bone formation (improvement)	No pain/no ONJ signs	Stable (improvement)	No pain/no ONJ signs
**#7**	**F** **E** **D**	**7**	II	Slightly improved bone formation (improvement)	No pain/no ONJ signs	n.a.	n.a.

Legend: A = abscess; P = pain; De = Mucosal dehiscence; E = exposed bone; F = fistula; Pa = paresthesia; Pu = purulent discharge; S = swelling. * according to Bedogni et al. [19]; n.a. = not applicable.

## Data Availability

Not applicable.

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
