# Peer review of "Ozone Infiltration for Osteonecrosis of the Jaw Therapy: A Case Series"

_jcm, 2022, doi:10.3390/jcm11185307_

Round 1
Reviewer 1 Report
Please clarify the role of examiner, data collector and operator in order to remove the possibility of bias.
Please add some more clinical figures even for all cases.
Author Response
Please clarify the role of examiner, data collector and operator in order to remove the possibility of bias.
R: As suggested, this role has been requested. On the basis of the review process, Dr. Maria Chiara Terranova is no more listed as a co-author, her valuable help is defined in the acknowledgment.
Please add some more clinical figures even for all cases.
R: Only one complete case was considered for publication since the images are of high resolution. Duplicating this resolution cannot be ensured. Please clarify if you wish us to send all cases (clinical and radiological images) for publication or for evaluation by the Reviewers.
Reviewer 2 Report
Thank you for the opportunity to evaluate your manuscript. I have some suggestions that I hope you will find helpful.
1. Title:
- the term "open clinical trial" may be misleading due to the expected larger study group - remove it;
- you report a series of cases in the manuscript - add this information in the title;
- correct these in the other sections as well.
2. Affiliations:
- please add addresses;
- do not use abbreviations.
3. Abstract:
- remove the term T0, which may be confusing to readers by its similarity with the TNM classification;
- instead of "long follow-up period" write a value or a time range;
- correct these in the other sections as well.
4. Introduction:
- expand the abbreviation "AAOMS";
- knowledge about MRONJ is changing dynamically - remove sources older than 5 years if not absolutely necessary;
- in general matters on the subject of MRONJ, try to base mainly on current review articles, and in the detailed topic of the discussed therapy on all types of articles;
- correct these in the other sections as well.
5. Materials and methods:
- use the subsections (with their names) throughout the manuscript, in accordance with the STROBE checklist;
- present the inclusion and exclusion criteria in the table;
- specify the stage of MRONJ in the inclusion criteria.
6. Discussion:
- the death of two of the patients during the follow-up period requires extensive commentary, including:
- could the therapy in question have contributed to death?
- did treatment of MRONJ in terminal patients not expose them to unnecessary suffering?
- when referring to sources, use numerical designations instead of authors' names and publication dates.
7. Conclusions:
- this section is too broad - conclusions should briefly answer the research question / hypothesis, so consider moving some or all of the content of the current conclusions to the discussion section and creating 2-4 short and clear sentences of the conclusions.
In my opinion, the manuscript is a research report important for the development of MRONJ treatment techniques and should be re-evaluated after corrections have been made.
Author Response
Thank you for the opportunity to evaluate your manuscript. I have some suggestions that I hope you will find helpful.
- Title:
- the term "open clinical trial" may be misleading due to the expected larger study group - remove it;
- you report a series of cases in the manuscript - add this information in the title;
- correct these in the other sections as well.
R: The title and text have been revised, as suggested.
- Affiliations:
- please add addresses;
- do not use abbreviations.
R: The address has been added and the abbreviations removed.
- Abstract:
- remove the term T0, which may be confusing to readers by its similarity with the TNM classification;
- instead of "long follow-up period" write a value or a time range;
- correct these in the other sections as well.
R: The T0 in the Abstract has been removed and the time range of the follow-up defined. The term T0 has been associated with “first visit” in the text for clarification.
- Introduction:
- expand the abbreviation "AAOMS";
- knowledge about MRONJ is changing dynamically - remove sources older than 5 years if not absolutely necessary;
- in general matters on the subject of MRONJ, try to base mainly on current review articles, and in the detailed topic of the discussed therapy on all types of articles;
- correct these in the other sections as well.
R: As suggested, the abbreviation AAOMS has been defined. Most of the references were published in the last 5 years. The older references are related to the application of ozone therapy in patients with MRONJ, thus it is important in our opinion to retain these references due to the study being proposed. Indeed, there are not many studies evaluating such MRONJ therapy.
- Materials and methods:
- use the subsections (with their names) throughout the manuscript, in accordance with the STROBE checklist;
- present the inclusion and exclusion criteria in the table;
- specify the stage of MRONJ in the inclusion criteria.
R: Subsections have been added throughout the manuscript, in accordance with the STROBE checklist. The inclusion and exclusion criteria have been moved to Table 1. Even if all MRONJ stages had been included, all the patients presented stage II of MRONJ.
- Discussion:
- the death of two of the patients during the follow-up period requires extensive commentary, including:
- could the therapy in question have contributed to death?
- did treatment of MRONJ in terminal patients not expose them to unnecessary suffering?
- when referring to sources, use numerical designations instead of authors' names and publication dates.
R: The death of the two patients was related to the progression of their primary disease (metastases of breast cancer and prostate cancer). The local application of ozone does not seem to be related to their death. The local application of ozone facilitated the bony sequestrum of the necrotic bone and its spontaneous expulsion, the latter improving the quality of life of the patients.
- Conclusions:
- this section is too broad - conclusions should briefly answer the research question / hypothesis, so consider moving some or all of the content of the current conclusions to the discussion section and creating 2-4 short and clear sentences of the conclusions.
R: Effected.
In my opinion, the manuscript is a research report important for the development of MRONJ treatment techniques and should be re-evaluated after corrections have been made.
R: We thank the Reviewer for these suggestions and comments.
Round 2
Reviewer 2 Report
Thank you kindly for the opportunity to re-evaluate the manuscript. The authors responded to all my comments and made corrections.
I believe that the authors should reconsider the MRONJ stages in the inclusion criteria. In my opinion, the assumption of including stage 0 patients in this study is controversial, and stage I still may be regarded by some clinicians as excessive treatment. I suspect that the study group consisting solely of stage II patients is the result of proper qualification, and the theoretical assumption of including patients at all stages is a mistake.
Author Response
Thank you kindly for the opportunity to re-evaluate the manuscript. The authors responded to all my comments and made corrections.
R: Thank you for all the comments and suggestions.
I believe that the authors should reconsider the MRONJ stages in the inclusion criteria. In my opinion, the assumption of including stage 0 patients in this study is controversial, and stage I still may be regarded by some clinicians as excessive treatment. I suspect that the study group consisting solely of stage II patients is the result of proper qualification, and the theoretical assumption of including patients at all stages is a mistake.
R: We agree with the Reviewer that the possible inclusion of stage 0 patients, according to the AAOMS staging system, may be controversial. However, we applied the clinical-radiological SICMF-SIPMO staging system (Campisi G, Fedele S, Fusco V, Pizzo G, Di Fede O, Bedogni A. Epidemiology, clinical manifestations, risk reduction and treatment strategies of jaw osteonecrosis in cancer patients exposed to antiresorptive agents. Future Oncol. 2014 Feb;10(2):257-75. doi: 10.2217/fon.13.211. PMID: 24490612), that does not include stage 0, since questionable. Please consider that all cases of our study are stage 2 but it happens randomly and not as inclusion criteria.
Additionally, we specified in table 1 that we were applying the SICMF-SIPMO staging system.